# Experimental Verification of Optimized Anatomies on a Serial Metamorphic Manipulator

**DOI:** 10.3390/s22030918

**Published:** 2022-01-25

**Authors:** Nikolaos Stravopodis, Charalampos Valsamos, Vassilis C. Moulianitis

**Affiliations:** Department of Products and Systems Engineering, University of Aegean, 81100 Ermoupolis, Syros, Greece; balsamos@syros.aegean.gr

**Keywords:** serial modular metamorphic manipulators, optimal anatomy determination, anatomy to task matching, kinematic tasks, dynamic tasks

## Abstract

The inherit complexity of the determination of the optimal anatomy and structure to task requirements and specification for metamorphic manipulators poses a significant challenge to the end user, as such methods and tools to undertake such processes are required for the implementation of metamorphic robots to real-life applications in various fields. In this work, the methodology for an offline process for the determination of the optimal anatomy maximizing performance under different requirements is presented. Such requirements considered in this work include the kinematic, kinetostatic and dynamic performance of the manipulator during task execution. The proposed methodology is then applied to a 3 D.o.F. metamorphic manipulator for different tasks. The presented results clearly show that a single metamorphic structure is able to provide the end user with different anatomies, each better suited to task specifications.

## 1. Introduction

As robots become increasingly incorporated in human life and activities, so are robotic tasks becoming increasingly complex and challenging. Since the beginning of their implementation, robots have been closely linked to manufacturing and industrial activities, where most tasks that were undertaken by their usage were relatively easy to model and plan. While these tasks were becoming increasingly complex, the challenge was addressed, utilizing advanced sensing and control techniques, to overcome certain limitations of the robot’s anatomy. However, as the task pool moved on to include a multitude of tasks from various sectors of application (such as medicine, space, human care, etc.), their complexity increased almost exponentially. Coupled with the need for high adaptability and high performance, it soon became evident that robot design was required to move from the typical fixed anatomy systems to implement the reconfigurability paradigm so as to provide the additional capability of matching the robot’s anatomy to a task in an optimal fashion, thereby addressing the new requirements.

The incorporation of modularity and reconfigurability allowed for the design and production of various proposed robotic manipulator systems that provided end users with a highly adaptable and cost-effective system, since a modular reconfigurable robot may be loosely characterized as a “multiple anatomy system”. The new design paradigm allowed the end user to structure the best anatomy to match a given task such that the manipulator can achieve the best possible performance during its execution. It is beyond the scope of this work to present the multitude of different proposed design approaches and the subsequent proposed systems implementing this design paradigm; however, thorough reviews with extensive information and system presentation may be found in [1,2,3].

Although modularity and reconfigurability have been attained in the relevant literature as a most promising approach, the efforts to overcome certain barriers still present in the current robot design have led to the emergence of a new design paradigm: that of metamorphosis. The resulting designs, i.e., metamorphic robots, present much higher adaptability and reconfigurability than either fixed anatomy robots or modular reconfigurable robots since they incorporate the best features of both designs. The most important feature of metamorphic robots is the segregation of the notions of structure and anatomy. Both for fixed and modular reconfigurable robots, these terms are essentially the same. Even for modular reconfigurable robots, a given structure/anatomy is considered to be different than another one, and the transition from the first to the second requires down times and disassembly and reassembly processes. Metamorphic robots utilize movable connectors to facilitate the alteration of a given structure to different anatomies without the requirement for any change in the structure [4,5,6,7,8,9,10]. Moreover, due to their movable connectors, metamorphic manipulators are not bounded with respect to the possible anatomies that a structure can be altered to, which are usually imposed upon the structure due to the available types of connectivity of the various modules that comprise it. As such, non-standard anatomies may be achieved, something not common with fixed anatomy or modular reconfigurable robots, which provide additional advantages for increased performance during task execution [5,10,11,12,13].

However, with respect to task planning and design, metamorphic manipulators present a far greater challenge as opposed to both fixed anatomy and modular reconfigurable robots. In the case of fixed anatomy robots, task planning for high performance usually incorporates the optimal placement of the task in the manipulator’s workspace or the determination of the optimal sequence of the task points to be reached by the end effector [14,15,16,17]. Further augmentation is usually achieved via the incorporation of advanced control schemes and advanced sensing techniques [18]. In the case of modular reconfigurable robots, the added element of determining the optimal anatomy of the robot for the task to be performed provides for a great increase in task planning complexity [19,20,21,22]. In the case of metamorphic manipulators, the process is much more complex since a metamorphic structure best befitting the task must be determined first, then the optimal anatomy for the given task must also be determined, and finally the task is to be optimally planned for the derived optimal structure and anatomy. As such, in most approaches, these elements of the design process are usually considered individually [4,5,7,8,9,10,11,23].

The present paper presents a methodology for the determination of the optimal anatomy of a 3 D.o.F. metamorphic manipulator under different kinematic and kinetostatic performance considerations and the subsequent evaluation of the derived anatomies with respect to expected performance in a metamorphic manipulator physical system. Given the envisaged application of metamorphic manipulators to different types of tasks, the task considered in this work is a simulation of the common upper limb rehabilitation task for patients under the effects of strokes, where the manipulator is used to “teach” the patient’s brain in the movement of the upper limb that has lost mobility.

The contribution of the present work is twofold. Firstly, it presents a physical evaluation of the determined optimal anatomies performance under the selected task and performance measures utilized. Most of the current literature on metamorphic robots bases the evaluation of the optimal derived anatomies to the results of computational simulations and as such, the examination of a physical model of the manipulator could provide increased insight on the level of performance enhancement achieved. Although computational simulations are quite acceptable in terms of results, they usually ignore certain aspects of the physical robot: for example, in the kinematic performance optimization of tasks, the dynamic aspects of the manipulator are generally ignored; however, they could significantly affect the actual robot performance. Secondly, the optimization process is such that it is conducted rapidly offline, allowing the rapid metamorphosis of the metamorphic manipulator to the optimal anatomy and, as such, reducing down times. Moreover, the subsequent utilization of both kinematic and kinetostatic measures also allows for a more balanced performance by the derived anatomies in terms of kinematic and dynamic performance.

The rest of the paper is structured as follows. In the Materials and Methods section, the theoretical aspects of the optimization process, along with the kinematic and kinetostatic performance measures utilized, and the subsequent generic task description are presented in detail. Additionally, the physical workcell utilized to conduct the experimental verification of the results is also presented in this section. The Results section provides a presentation of the experimental results of the presented methodology, along with their respective interpretation. Finally, the paper closes, presenting the subsequent conclusions and the presentation of the envisaged future work to be undertaken into the subject by the authors.

## 2. Materials and Methods

### 2.1. Serial Modular Metamorphic Manipulators

The serial modular metamorphic manipulator (SMM) concept was first introduced in [4,5] as a hybrid class of robotic manipulators that retained advantageous characteristics of both modular reconfigurable and fixed anatomy robots. Making use of 1 D.o.F. reconfigurable connecting modules called “*pseudo joints*”, it allowed a single structure composed of such modules, active modules (robot joints) and passive modules to be metamorphosed to different anatomies. As such, the end user was attributed to a robotic system that allowed its rapid assembly or reassembly to a structure best befitting their needs, being able to attain a multitude of different anatomies best matching the required task and performance characteristics. An example of a 3 D.o.F. metamorphic structure composed of three active joints and two pseudo joints, as well as a 3D model of the pseudo joint connector are presented in Figure 1. In Figure 1a, the respective twists of the active joints and pseudo joints of the presented structure are illustrated as ξ_1_,ξ_2_,ξ_3_,ξ_a_,ξ_β_, respectively.

The pseudo joint module presents a single rotational D.o.F. that allows the relative rotation of its two parts, thus subsequently altering the relative Denavit–Hartenberg parameters of the manipulator with each different configuration. As such, the current anatomy of an SMM structure may be implicitly defined by the current configuration of the pseudo joints in its lattice, depicted by the vector Q_a_ of the angles of the pseudo joints, termed, therefore, as the metamorphic parameters of the manipulator. During operation, these values remain unchanged. As such, a modular parametric analytical solution as a function of the metamorphic parameters for the kinematics and dynamics of each structure is also able to be structured. In [4,5], the analytic formulation of the forward and inverse kinematics of this manipulator class as well as the determination of a structure’s Jacobian were generated through the use of the Screw theory and the product of exponentials (POE) formula.

As already presented, one of the key issues regarding the optimal task planning for metamorphic robots is the determination of the optimal anatomy for the task considered, i.e., the matching of possible anatomies to task. In this work, the metamorphic structure is considered to have been a priori determined, and the method presented aims for the determination of the best possible anatomy for the considered task. The determination is conducted under selected kinematic and kinetostatic performance aspects of the manipulator. Therefore, it is important to present the task formulation and the selected performance measures utilized in this work before the presentation of the selected method.

#### 2.1.1. Task Definition

In general, most robotic tasks may be considered as trajectory following tasks, where the end effector of the manipulator is to follow a prescribed trajectory while executing different types of processes at given points along the trajectory or on its whole. Application examples of this consideration are a pick and place task, such as in assembly applications requiring the manipulator to grasp and release objects at different sites in its workspace, but the end effector does follow a trajectory during the motion between task points; during spot welding, the end effector has to follow a given trajectory between the different task points; in the case of the placement of glue or arc welding of surfaces, the end effector has to follow the trajectory that will allow the application on the processed surfaces, etc. Such trajectories may be 2D or 3D, depending on the task and the workcell structure. However, in every case, a trajectory may be defined as a collection of points in the manipulator’s workspace with a predefined specified sequence and respective placement with respect to each other. The placement of the points can be unilaterally determined by the placement of local coordinate systems at each one, a process that allows for two considerations: (a) the definition of the respective position and orientation between points and, as such, of the whole trajectory relative to a base coordinate system (usually the robot’s base in the case of fixed manipulators) and (b) the a priori placement of these frames so that they depict the required end effector position and orientation at the trajectory points. Therefore, a robotic task may be defined as a collection of task frames along a given trajectory, depicting the position and orientation of the end effector at each trajectory point during task execution. A mathematical description of a robotic task can therefore be given as
(1)$$Tr=\left\{P_{i}\right\},\;i=1,2,\ldots ,n$$
where *n* is the number of frames that the trajectory consists of and
(2)$$P_{i}=\left[\begin{array}{cc} \;R_{i} & p_{i}\\ 0_{1\times 3} & 1 \end{array}\;\right]$$
where $\;R_{i}$ is the 3 × 3 orientation matrix of the task frame relative to the base frame of the manipulator depicting the orientation of the end effector when at the *i*^th^ point, and *p_i_* is the coordinated task frame relative to the base frame of the manipulator.

#### 2.1.2. Selected Local Performance Measures

Most performance measures, whether kinematic, kinetostatic or dynamic, are local measures, i.e., quantifying the manipulator’s performance at a given point in its workspace. Additionally, each performance measure quantifies a selected aspect of the manipulator’s performance. As such, different results may occur when considering different performance aspects of the manipulator’s performance. In the presented work, three local performance measures were considered.

Manipulator velocity ratio (MVR)

The MVR was first proposed by Dubey and Luh in [24] and provides a measure of the quality of the transmission of the joint speeds to the end effector velocity at a given point in the manipulator’s workspace and at a given direction of motion. The index is the ratio of the end effector velocity to the required joint speeds to achieve it along the given direction of motion. The mathematical expression of the measure is
(3)$$r_{v}\left(q\right)=\frac{1}{\sqrt{u_{v}^{T}\left(J_{v}\left(q\right)J_{v}^{T}\left(q\right)\right)u_{v}}}$$
where $u_{v}$ is a unit vector in the direction of motion of the end effector, $J_{v}$ is the manipulator’s kinematic Jacobian, and q is the manipulator’s joint variables vector depicting its current configuration.

Manipulator mechanical advantage (MMA)

Similar to the MVR, the MMA was introduced in [24] and is a kinetostatic local performance measure depicting the quality of transmission of joint torques to the end effector forces applied at a given direction, depicting the ratio of the end effector applied force in a given direction to the required joint torques to achieve it. The mathematical expression of the measure is
(4)$$r_{m}\left(q\right)=\frac{1}{\sqrt{u_{m}^{T}\left(J_{m}\left(q\right)J_{m}^{T}\left(q\right)\right)u_{m}}}$$
where $u_{m}$ is a unit vector in the direction of the application of force by the end effector, $J_{m}$ is the manipulator’s dynamic Jacobian, and q is the manipulator’s joint variables vector depicting its current configuration.

Manipulator’s effective mass (MU)

The effective mass measure was proposed by Khatib in [25] and is a measure of the component of linear acceleration along a given direction of motion for the application of a unit force applied along this direction. The mathematical expression of the measure is
(5)$$\frac{1}{m_{u}\left(\Lambda_{u}\left(q\right)\right)}=u^{T}\Lambda_{u}^{-1}\left(q\right)u$$
where *u* is the unit vector in the direction of the application of force, *q* is the manipulator’s joint variables vector depicting its current configuration, and $\Lambda_{u}^{-1}\left(q\right)$ is the pseudo kinetic energy matrix, which characterizes the translational response of the end effector to an applied force and is given as
(6)$$\Lambda_{u}^{-1}\left(q\right)=J_{v}\left(q\right)A^{-1}\left(q\right)J_{v}^{T}\left(q\right)$$
where $A^{-1}\left(q\right)$, is the inverse inertia matrix and for the metamorphic structure under study can be calculated as presented in [26].

#### 2.1.3. Structuring of Task-Based Performance Measures

Given the fact that the selected performance measures are (a) local and (b) dependent on the current manipulator posture (configuration) depicted by the vector *q*, structuring the required task-based measures requires the determination of an expression of an overall measure value along the task (trajectory) and the incorporation of the metamorphic manipulator anatomical values. As such, the proposed task-based measures are the following:Average MVR along the task
(7)$$\underline{r_{v}}\left(q_{a}\right)=Average\left(\left\{r_{v}\left(q_{a},q_{i}\right):q_{i}\;\to \;P_{i}\right\},\;i=1,2,\ldots ,n\right)$$

Average MMA along the task


(8)
$$\underline{r_{m}}\left(q_{a}\right)=Average\left(\left\{r_{m}\left(q_{a},q_{i}\right):q_{i}\;\to \;P_{i}\right\},\;i=1,2,\ldots ,n\right)$$


Average MU along the task

(9)$$\underline{m_{u}\left(\Lambda_{u}\left(q_{a}\right)\right)}=Average\left(\left\{m_{u}\left(\Lambda_{u}\left(q_{a},q_{i}\right)\right):q_{i}\;\to \;P_{i}\right\},\;i=1,2,\ldots n\right)$$
where in the above, $q_{a}$ is the vector of metamorphic variables of the metamorphic manipulator depicting a given anatomy, and *q_i_* is the configuration of the anatomy $q_{a}$ at each task point $P_{i}$. A large number of intermediate path points is generated for the evaluation of the local index values. However, since what is sought with respect to the determination of the optimal anatomy is information that can be attributed to the whole path, in a compensating manner, the average value of the induces along the path is selected.

### 2.2. Methodology for the Determination of the Optimal Anatomy for a Given Task under the Proposed Performance Measures

The proposed methodology is graphically presented in Figure 2.

The inputs of the method presented are the metamorphic manipulator’s structure which is considered to be a priori determined and the task description as the set of task frames to be visited by the end effector with a given sequence and orientation, as presented in Equation (1).

Initially the method evaluates all possible anatomies of the given metamorphic structure depicted by the set
Q_a_ = {*q_a,j_*}(10)
where *j* = 1, 2, …, m are the possible anatomies that the current structure may be metamorphosed to. Anatomies that may present collision possibilities between the manipulator’s modules in the structure are removed from the available anatomies set, using the procedure presented in [27]. At the end of this process, the set of feasible anatomies Q_a,feasible_ is determined. A resolved motion rate control algorithm [28] is then used to determine the required configurations for task execution by each anatomy, resulting in the set of required configurations of the anatomy {q_i_(Q_a_feasible_)}. The integration step utilized is very small in order to produce a large number of intermediate frames along the path. At each calculation, the Jacobian determinant is evaluated and if it is smaller than a set threshold, the examined anatomy is eliminated so as to avoid passing near singular points during task execution. At each intermediate frame, the selected index value is also evaluated.

A subsequent anatomy elimination process is then conducted, utilizing the joint motion bounds q_i_bounds_ that are considered for the task execution. Anatomies for which the determined *q_i_* is out of bounds are subsequently removed from the pool. If the evaluated anatomy successfully completes the path, the average values of the selected indices are determined (Equations (7)–(9)). For each of the performance indices, an optimal anatomy is extracted.

In order to extract these optimal anatomies, an exhaustive search of the indices is conducted, and the anatomies with the best overall index values are selected. Afterwards, for each of the optimal anatomies, the corresponding trajectory frames are retrieved and are used for the physical task implementation to the laboratory prototype. The procedure presented for the determination of the optimal anatomies is conducted offline due to the required time and computational resources.

The presented steps of the proposed process were selected in order to avoid an unfeasible solution, i.e., the derivation of an optimal anatomy that would not be physically able to perform the task. However, the derivation of the optimal anatomy is dependent on the task specification and the selected indices.

### 2.3. Experimental Metamorphic Manipulator Workcell

The proposed method was applied to the experimental metamorphic manipulator workcell developed at the Dept. of Product and System Design Engineering at the University of Aegean. The following paragraphs provide a description of the workcell and the tasks to which the proposed method was applied.

#### 2.3.1. Serial Metamorphic Manipulator Prototype

The serial metamorphic manipulator (SMM) prototype, presented in Figure 1a (CAD version) and in Figure 3 (physical system), was built for the experimental validation of the theoretical results extracted, regarding the optimized kinematic [5] and dynamic [10,26] properties of this class of reconfigurable manipulators.

The first joint is fixed to the robot base, and its twist (ξ_1_) is parallel to the robot’s base +z axis. A hybrid stepper motor coupled with a planetary gearbox of high gear ratio is selected to provide the necessary torque for manipulator operation. The rest of the robot structure is modular, with two types of actuators used. For active operation, the Dynamixel PH All-in-One Industrialized Smart Servo actuators are used [29], while for system autonomous reconfiguration, prototype passive modules (pseudojoints) are designed and built [30]. The latter are 1 D.o.F. (rotational), offline operated modules used to form the metamorphic links of the manipulator. Once the manipulator is assembled at a certain structure, the user can change the angle of each passive module, and a wide range of robot anatomies emerges. The passive modules can accomplish discrete reconfiguration for a total of 15 angles, and the vector of the metamorphic variables *q_a_* = [*q_a_*_1_ *q_a_*_2_]^T^ is used to denote the current setting of the pseudojoints’ angles. The main torque transmission mechanism of the passive modules consists of a worm-type gearbox, which is a self-locking mechanism, so there is no need of a mechanical or dynamic braking system. After anatomy metamorphosis is complete, the modules can be screw-locked, and no power is consumed when the active operation mode (AOM) is triggered. Possible structure assemblies derive from the total number of passive modules used and the available connectivity surfaces of both modules. The Dynamixel actuators provide a single input and a single output connection frame, while the passive modules provide one output and two input surfaces. For each surface, multiple positions for screw connections exist, leading to numerous possible assemblies.

The robot prototype operation architecture is presented in Figure 4. An Arduino DUE microcontroller is used for AOM implementation. Communication between the controller board and the PC is implemented through the universal asynchronous receiver/transmitter (UART) protocol. The manipulator AOM firmware is uploaded on the microcontroller, and user input/robot output are transmitted/received using the higher-level software libraries provided and the serial monitor built in the Arduino IDE. During AOM, the pseudojoints are powered down and locked using screw bolts. For POM, an Arduino MEGA micro-processor is used as a master device which communicates with the pseudojoints through the serial peripheral interface (SPI), synchronous serial communication protocol. To activate/deactivate each operation mode, the user selects the specified serial port of each microcontroller in the PC and sends the corresponding signal. Since the two microcontrollers cannot communicate with each other to identify the current robot anatomy, the user should always verify that the motion commands given during AOM are extracted for the current anatomy. For task space operation of the SMM prototype, the joint commands are first offline computed, utilizing the kinematic and dynamic libraries developed in MATLAB. Each trajectory task is generated on the user PC, and the joint motion commands (position, velocity, acceleration, execution time) are loaded on a header file that is sourced from the robot’s AOM firmware.

During task execution, the encoders integrated in active modules are used to extract the current position data. For accessing the current velocity data, an indirect method is applied. Since a brushless DC (BLDC) motor is integrated in the actuator, an inverter is used in order to control the motor speed through the PWM signal produced by the position and velocity control system. Given the analog pin value of the MCU, the average voltage generated to power the motor is measured, and the motor speed is estimated given the motor rated speed characteristic curve. This process is internally implemented in the actuator’s firmware, and information is extracted by the user through higher-level system functions and commands. The hybrid stepper motor of the first active joint has an encoder that also provides the current robot position, but the motor velocity is estimated given the step pulse width generated in order to move the motor. Since the step angle of the motor is fixed, the instantaneous angular velocity can be calculated given the time width of the step pulse command, generated by the MCU. All joints are equipped with current sensors in order to measure the current drawn by the joints’ motors.

For safety measures, the first active joint is equipped with position limit switches and a current sensor. If the switches are triggered or a current above the peak value set for the hybrid stepper motor driver is sensed during active operation, the robot controller stops the robot motion. A manual-triggered emergency button is also integrated in the main robot power supply line. The Dynamixel actuators also provide the ability of setting joints’ limit angle and current values. Moreover, extra safety is guaranteed in the case of overload, electrical shock and overheating errors. If such an instance occurs, motor shutdown occurs, the dynamic brake is activated and the motor stops.

#### 2.3.2. Trajectory Generation and Implementation

In this work, a single robot structure is evaluated. The structure initial assembly is presented in Figure 1a. It consists of two passive modules, which are both connected using the same connectivity surface (2) presented in Figure 1b. The joint limits for position, velocity, acceleration and torque are given in Table 1. A mechanical gripper is used for the attachment of loads. A 3-axis force sensor is also placed between the last active joint and the end effector. In the experiments conducted in this work, only sensor measurements (position/velocity/current) from the active joints are sampled.

The path-generation procedure presented in Figure 5 is executed offline. First, the anatomies that will be further evaluated should be produced. For the defined SMM structure, the set of all possible anatomies is exhaustively investigated in order to distinguish anatomies with a serial chain that contains colliding bodies. The anatomies found are removed from the final set of the feasible anatomies, which is exhaustively investigated to obtain the best-scoring anatomies for the task implementation.

The initial point for each task execution is defined in joint space. It should be highlighted that for each set of pseudojoint angles, a different point in the task space is produced. The task is defined in task space, as the velocity twist of the end effector. In Table 2, the initial point and the end-effector velocity for each of the executed tasks are presented.

In order to produce the task waypoints in joint space, the resolved motion rate control (RMRC) defined in [28] is implemented using the methodology described in [31]. For each anatomy, the extracted joint angles are evaluated given the joint position bounds of the physical robot. In order to avoid kinematic singularities, a minimum value for the determinant of the Jacobian matrix is defined, and the necessary condition is examined at each integration step. If all calculated waypoints satisfy the joint bound constraints and no singularities occur along the calculated trajectory points, the generated path and the corresponding anatomy are saved. For each of the saved anatomies/trajectories, the performance indices presented in Section 2.2 are calculated along the trajectory points, and an average value is extracted. The charts of the variation of the performance indices for each anatomy are compared, and the best scoring anatomy for each index can be later selected for task implementation on the SMM prototype.

The overall procedure of task implementation, on the SMM prototype, is presented in Figure 6. First, the procedure previously presented is executed offline. This results in a database of the extracted trajectory waypoints for all feasible and successfully simulated anatomies. Then, the user can select the best-scoring anatomies of each index and implement the task on the SMM prototype currently located in the laboratory. For each task, first the anatomy selected should be reached on the real robot. This is offline executed during POM. When both pseudojoints are set to the corresponding angles, the user activates AOM. Then, the user can online select the trajectory path that matches the investigated anatomy and can also determine the actuator gains and the time duration of the segments. Here, equal time segments and a fixed proportional gain for the actuator controller are selected. A 0.5 kg load mass is attached on the end effector, and the joint position, velocity and current measurements are sampled during task execution. When the robot motion has finished, a new anatomy can be selected, and the overall procedure can be re-executed.

## 3. Results

### 3.1. Task 1: Linear Path (Direction Axis: +X, Distance: 20 cm)

The results of the application of the proposed method for a linear motion task of the end effector in the direction of the global *x*-axis with a distance of 20 cm for the considered kinematic and kinetostatic measures are presented in Figure 7.

As seen, the anatomies of the 3 D.o.F. manipulator that were finally considered after the elimination process were 10 in total. The best performing anatomy under the effective mass index (Figure 7a) was anatomy no. 38; the best performing anatomy under the MMA index (Figure 7b) was anatomy no. 31; and the best performing anatomy under the MVR index (Figure 7c) was anatomy no. 59. First and foremost, these results show the adaptability of the metamorphic manipulator to different task specifications, even for a given metamorphic structure, as the end user is able to alter the structure’s anatomy in order to enhance the manipulator’s performance for the task, based on the selected performance aspect. Similarly, examining the manipulator’s performance for the three derived anatomies, the graphs presented in Figure 8 are considered.

As illustrated in Figure 8, the resulting best anatomies present the expected performance behavior under each performance index. The anatomy that presents the highest MVR value utilizes the minimum possible joint velocities for the generation of the required end effector velocity along the path (Figure 8a), while the anatomy presenting the best average MMA value requires the minimum joint current usage (Figure 8b) for the end effector to apply the required force along the path. The anatomy presenting the best effective mass mean value utilizes joint speeds and requires joint currents that, for most of the path, remain between the values derived for the anatomies determined by the other two indices. However, the reduced joint current usage (as opposed to that of the MVR), signifies that the manipulator is able to exert larger accelerations of its joints along the path with reduced joint torque utilization. The increased current usage in the case of the optimal anatomy with respect to the MVR is due to the much larger inertia that joints 1 and 2 have to compensate for during motion, since this anatomy is quite extended.

Moreover, it was considered advantageous to compare anatomies performance under the same index, i.e., the best derived anatomy and the worst performing anatomy. For reference, the results for the MMA index are presented in Figure 9, while the respective results regarding the best and worst performing anatomies with respect to the effective mass measure are presented in Figure 10.

The results presented in Figure 9 and Figure 10 were expected, given the theoretical interpretation of the considered measures, and signify the validity of the presented method with respect to the derivation of the optimal anatomy. As seen in Figure 9, the best anatomy with respect to the MMA requires minimum joint current utilization in order for the end effector to apply the required force along the path as opposed to the worst performing anatomy under the same measure. Moreover, as depicted in Figure 10, the best derived anatomy with respect to the efficient mass measure requires a similar amount of joint current utilization to achieve higher joint velocities, i.e., acquiring larger joint accelerations along the path.

Finally, the optimal anatomies of the experimental metamorphic 3 D.o.F. manipulator with respect to each measure are presented in Figure 11 and Figure 12.

As seen in Figure 11, Figure 12 and Figure 13, the anatomies derived under each measure are significantly different and, moreover, are not typically met in current fixed anatomy manipulators.

### 3.2. Task 2: Linear Path (Direction Axis: −Z, Distance: 20 cm)

The results of the application of the proposed method for a linear motion task of the end effector in the direction of the global z-axis with a distance of 20 cm for the considered kinematic and kinetostatic measures are presented in Figure 14.

With respect to the results of the previous task, it can be seen in Figure 14 that the best performing anatomy under the effective mass index (Figure 14a) and the best performing anatomy under the MMA index (Figure 14b) was anatomy no. 12, while the best performing anatomy under the MVR index (Figure 14c) was anatomy no. 44. The similar performance behavior for the effective mass and MMA measures stems from the fact that the end effector applied force direction coincides for this particular task with the motion direction along the path. Similarly, examining the manipulator’s performance for the three derived anatomies, the graphs presented in Figure 15 are considered.

In accordance with the observations regarding the manipulator’s performance in the previous task, it can be seen in Figure 15 that the best anatomy for the MVR measure utilizes the minimum possible joint velocities for the generation of the required end effector velocity along the path (Figure 15a), while the anatomy presenting the best average MMA and effective mass value requires the minimum joint current usage (Figure 15b) for the end effector to apply the required force along the path.

The comparison of the best and worst performing anatomies with respect to each measure are subsequently presented in Figure 16.

The results presented in Figure 16 are analogous to those presented in the previous task and accordingly lead to the same conclusions.

Finally, the optimal anatomies of the experimental metamorphic 3 D.o.F. manipulator with respect to each measure are presented in Figure 17 and Figure 18.

As seen in Figure 17 and Figure 18, the anatomies derived under each measure are significantly different and, moreover, are not typically met in current fixed anatomy manipulators.

### 3.3. Task 3: Circular Path (Rotation Axis: +Z, Diameter: 15 cm)

The results of the application of the proposed method for a motion task of the end effector in circular motion with a Diameter of 15 cm for the considered kinematic and kinetostatic measures are presented in Figure 19. For this particular task, the MVR index was not considered due to limits beset on the first pseudo-joint angle.

With respect to the results of the previous task, it can be seen in Figure 19 that the best performing anatomy under the effective mass index (Figure 14a) was anatomy no. 31 and the best performing anatomy under the MMA index (Figure 14b) was anatomy no. 18. The manipulator’s performance for the two derived anatomies is presented in Figure 20.

As seen in Figure 20, both best derived anatomies present a rather similar behavior with respect to joint current utilization (Figure 20b), while the best anatomy derived under the effective mass presents the lower utilized joint velocities along the task (Figure 20a).

Finally, the optimal anatomies of the experimental metamorphic 3 D.o.F. manipulator with respect to each measure are presented in Figure 21 and Figure 22.

As seen in Figure 21 and Figure 22, the anatomies derived under each measure are significantly different and, moreover, are not typically met in current fixed anatomy manipulators.

### 3.4. Task 4: Circular Path (Rotation Axis: +Z, Diameter: 10cm)

The results of the application of the proposed method for a motion task of the end effector in circular motion with a diameter of 10 cm for the considered kinematic and kinetostatic measures are presented in Figure 23.

With respect to the results of the previous task, it can be seen in Figure 23 that the best performing anatomy under the effective mass index (Figure 23a) was anatomy no. 22, while the best performing anatomy under the MMA index (Figure 23b) and the best performing anatomy under the MVR index (Figure 23c) was anatomy no. 43. The similar performance behavior for the MVR and MMA measures stems from the fact that the end effector applied force direction coincides for this particular task with the motion direction along the path. Similarly, examining the manipulator’s performance for the three derived anatomies, the graphs presented in Figure 24 are considered.

As illustrated in Figure 24, the resulting best anatomies present the expected performance behavior under each performance index. The anatomy that presents the highest MVR and MMA value utilizes minimum joint velocities for the generation of the required end effector velocity along the path (Figure 8a) and, at the same time, minimum joint current utilization (Figure 8b) for the end effector to apply the required force along the path. The anatomy presenting the best effective mass mean value utilizes joint speeds and requires a joint current that, for most of the path, remain between the values derived for the anatomies determined by the other two indices. However, the reduced joint current usage signifies that the manipulator is able to exert larger accelerations of its joints along the path with reduced joint torque utilization.

The respective graphs comparing the performance of the best and worst performing anatomies for each measure are presented in Figure 25 and Figure 26.

The results presented in Figure 25 and Figure 26 were expected, given the theorical interpretation of the considered measures, and signify the validity of the presented method with respect to the derivation of the optimal anatomy. As seen in Figure 25, the best anatomy with respect to the MMA and the MVR requires minimum joint current utilization in order for the end effector to apply the required force along the path, and minimum joint velocities require for the end effector to achieve the required velocity along the path, as opposed to the worst performing anatomy under the same measure. Moreover, as depicted in Figure 26, the best derived anatomy with respect to the efficient mass measure requires a similar amount of joint current utilization to achieve higher joint velocities, i.e., acquiring larger joint accelerations along the path.

Finally, the optimal anatomies of the experimental metamorphic 3 D.o.F. manipulator with respect to each measure are presented in Figure 27 and Figure 28.

As seen in Figure 27 and Figure 28, the anatomies derived under each measure are significantly different and, moreover, are not typically met in current fixed anatomy manipulators.

## 4. Discussion

One of the key aspects and gains for the end user envisaged by the utilization of metamorphic manipulators is the rapid adaptation of a metamorphic structure to an anatomy best suited to the task parameters and presenting augmented performance with respect to task execution. Given the multitude of performance measures aiming to quantify and optimize performance under selected aspects of the manipulator’s operation, the variability of the anatomy of metamorphic structures could present a much sought bonus, especially regarding the implementation of such systems by SMEs, where capital investment opportunities and/or capabilities may be reduced or limited.

With respect to the utilization of such systems, two key issues are required to be addressed: first and foremost, the design and development of physical systems, and secondly, the development of the tolls and methods for the optimal task planning and structure and anatomy derivation for the maximization of the performance of the metamorphic manipulators.

The present research and the subsequently presented results provide contributions with respect to the second key aspect. The proposed methodology for the determination of the optimal anatomy with respect to task performance provides a rapid anatomy determination tool and is closely linked to the performance of the physical system. The applied metrics or measures that quantify the system performance presented may be extended easily to encompass other performance measures that are abound in the relative literature. Additionally, the presented results provide evidence that the offline simulation results do indeed lead to an increase in the manipulator’s performance, with respect to the selected aspect, whether it is a kinematic, kinetostatic or dynamic one.

Results show that in all cases, the 3 D.o.F. metamorphic manipulator presents a number of different anatomies that can provide consistently high performance under the selected performance indices. Given the task specifications, these anatomies may present similar performance under different performance measures, as in the case of Task 4, where the same optimal anatomy was determined under the MMA and effective mass indices. By utilizing the results for each task, the user is capable of selecting the optimal anatomy that best serves their purpose. For example, regarding task 1, the end used may choose to utilize the optimal anatomy with respect to the MVR index, where the robot utilizes the increased current for its motion, but allowing the end user to increase joint speeds during task execution, thereby shortening cycle times. On the other hand, the section of the optimal anatomy under the MMA or the effective mass indices will allow for lower current utilization and the application of greater forces by the end effector along the path. Similar remarks may be concluded from the results of all considered tasks. In general, anatomies derived under the proposed task-based kinematic index (average MVR value along the path) were found to be most “extended”, while those derived under the proposed dynamic and kinetostatic measures (effective mass and MMA, respectively) were found to be most “contracted”. Such a result was, in fact, expected since an extended mechanism is able to achieve higher end velocities, while a contracted one is able to exert larger forces.

## 5. Conclusions

A methodology for the determination of the optimal anatomy of a metamorphic manipulator under different performance measures was presented. The results of the offline process were utilized to conduct an experiment in order to validate the simulation results for a 3 D.o.F. metamorphic manipulator system. The experimental results showed that the derived best performing anatomy of the manipulator under each task did in fact perform the given task under consistent high performance, with respect to the chosen performance aspect. In the case of kinematic tasks, the optimal anatomy required significantly lower joint speeds to achieve the required end effector velocity along the task, while in the case of kinetostatic tasks, the manipulator utilized lower joint torques in order for its end effector to apply the required force along the task. On average, the derived optimal anatomies in the first case presented a 40%–45% reduction in utilized joint speeds during task execution, while in the case of the dynamic tasks, a similar reduction of about 30%–35% in current usage was determined. Moreover, it was also shown that a single metamorphic structure presents anatomies that could satisfy performance and operational requirements for different tasks, signifying the importance of metamorphic manipulator systems for future applications.

## Figures and Tables

**Figure 1 sensors-22-00918-f001:**
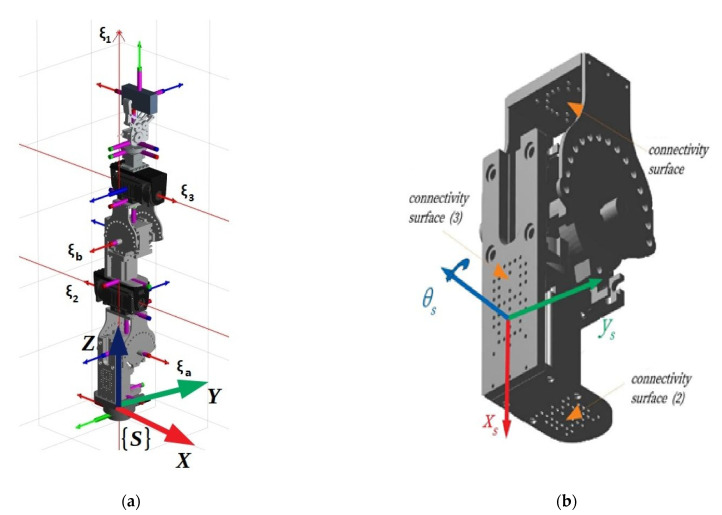
A CAD view of the SMM: (**a**) The structure under study; (**b**) the pseudo joint module and its respective connectivity surfaces.

**Figure 2 sensors-22-00918-f002:**
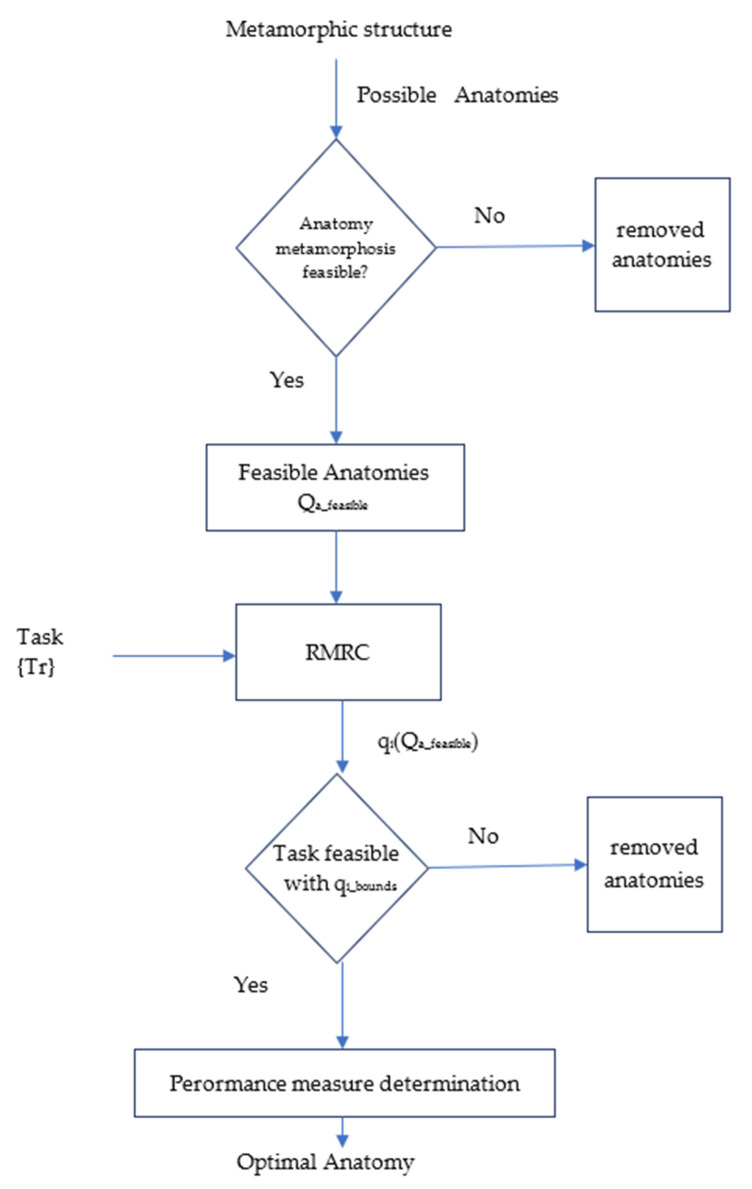
Flowchart of the proposed methodology.

**Figure 3 sensors-22-00918-f003:**
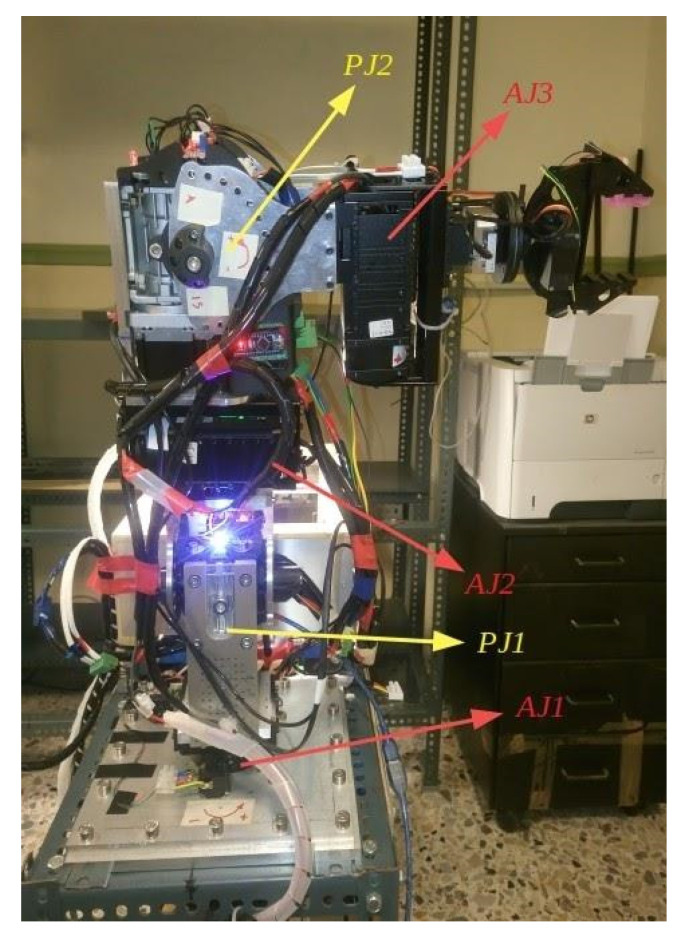
The SMM prototype used in this research. (All AJ and PJ1 are at zero position, and PJ2 is set to −1.5708 rad.)

**Figure 4 sensors-22-00918-f004:**
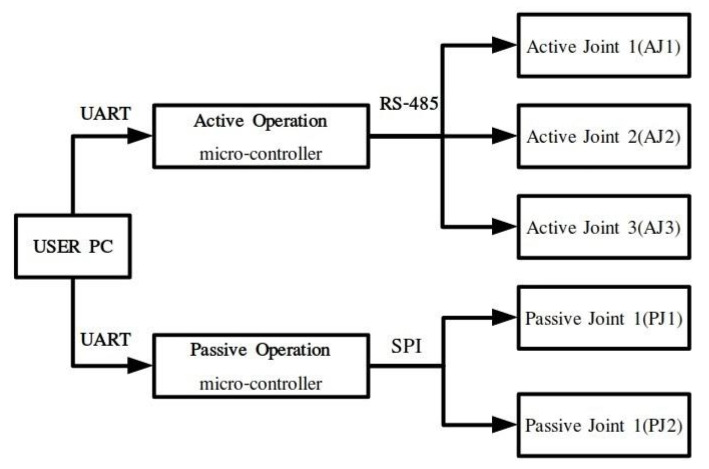
SMM system architecture.

**Figure 5 sensors-22-00918-f005:**
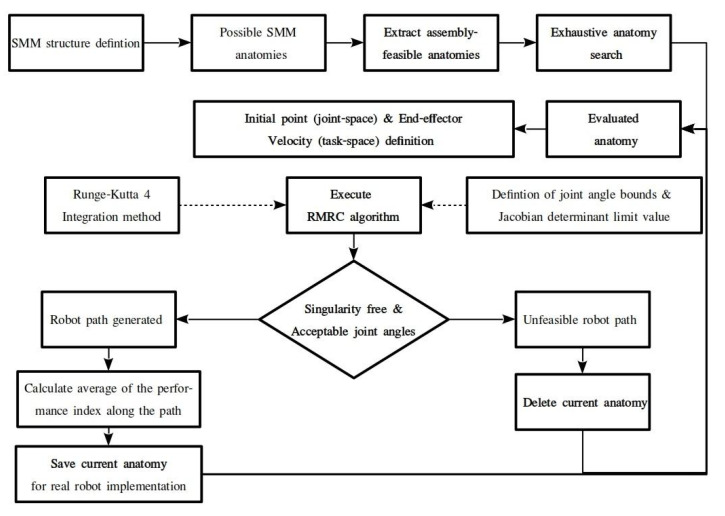
Database generation of trajectories for feasible anatomies of a SMM.

**Figure 6 sensors-22-00918-f006:**
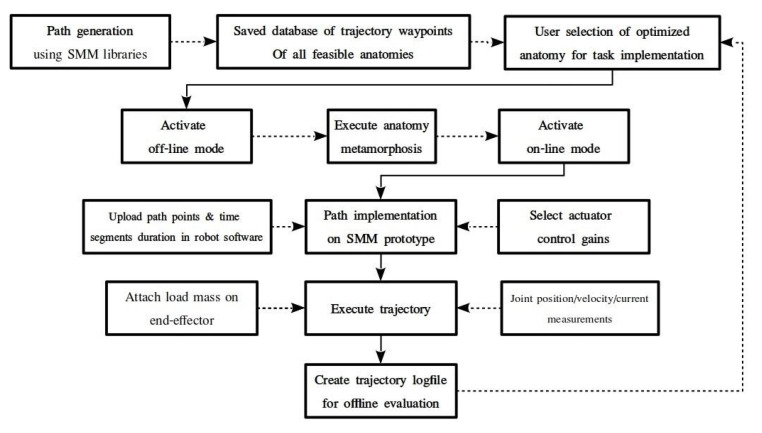
Task implementation on SMM prototype for a user specified optimized anatomy.

**Figure 7 sensors-22-00918-f007:**
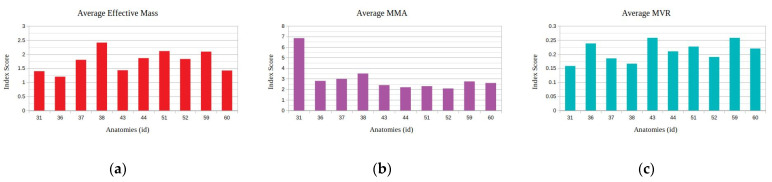
Derived mean index value for the selected performance indices under the available metamorphic structure anatomies, for Task 1 ((**a**) with respect to the Average Effective mass index, (**b**) with respect to the Average MMA index and (**c**) with respect to the average MVR index). The *y*-axis in all graphs represents the value of the according index.

**Figure 8 sensors-22-00918-f008:**
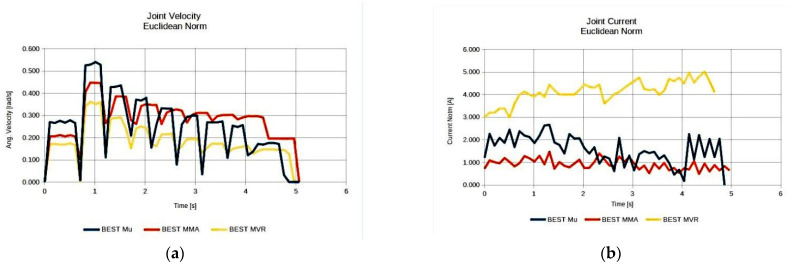
Comparison of the required joint velocities (**a**) and joint current usage (**b**) for the best anatomies under each performance index for Task 1.

**Figure 9 sensors-22-00918-f009:**
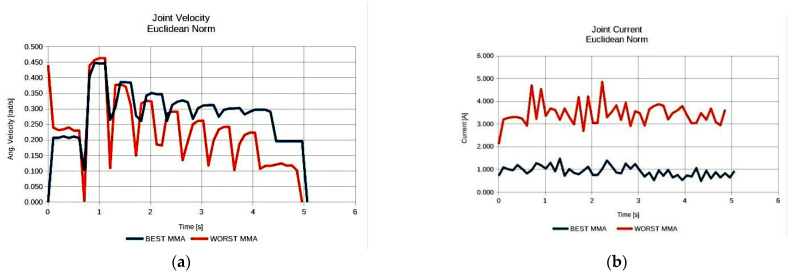
Comparison of the required joint velocities (**a**) and joint current usage (**b**) for the best anatomy and the worst performing anatomy under the MMA performance index for task 1.

**Figure 10 sensors-22-00918-f010:**
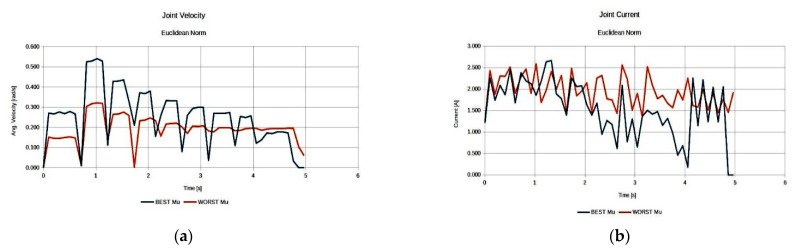
Comparison of the required joint velocities (**a**) and joint current usage (**b**) for the best anatomy and the worst performing anatomy under the effective mass performance index for task 1.

**Figure 11 sensors-22-00918-f011:**
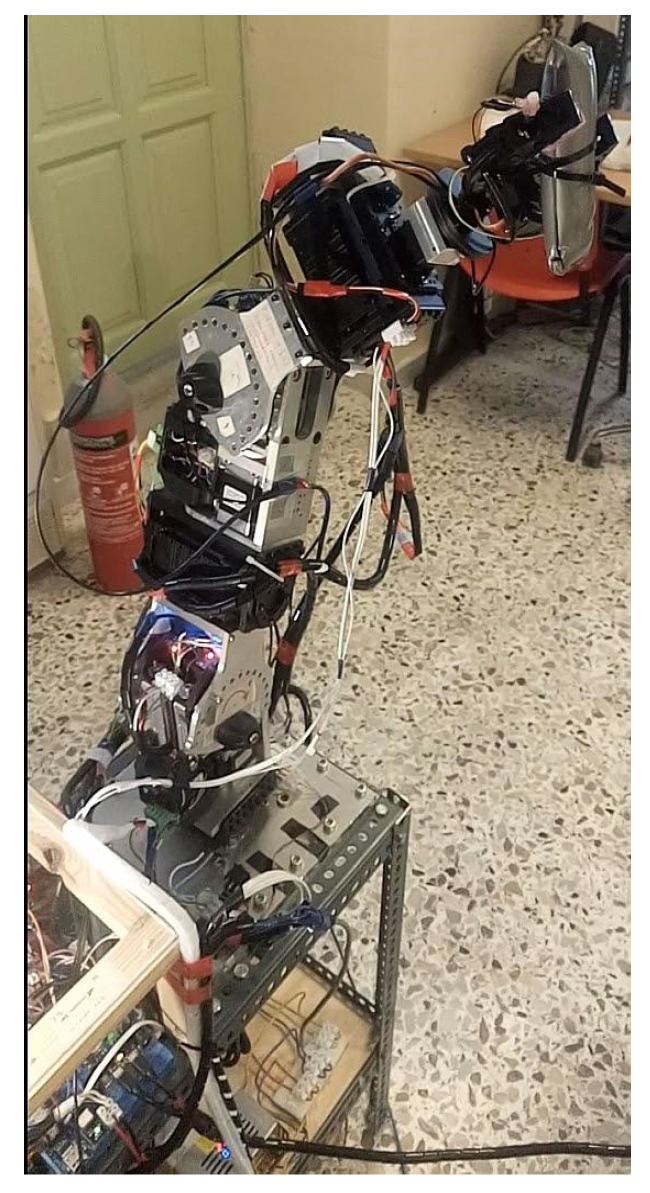
Optimal anatomy of the metamorphic manipulator derived under the effective mass measure for Task 1.

**Figure 12 sensors-22-00918-f012:**
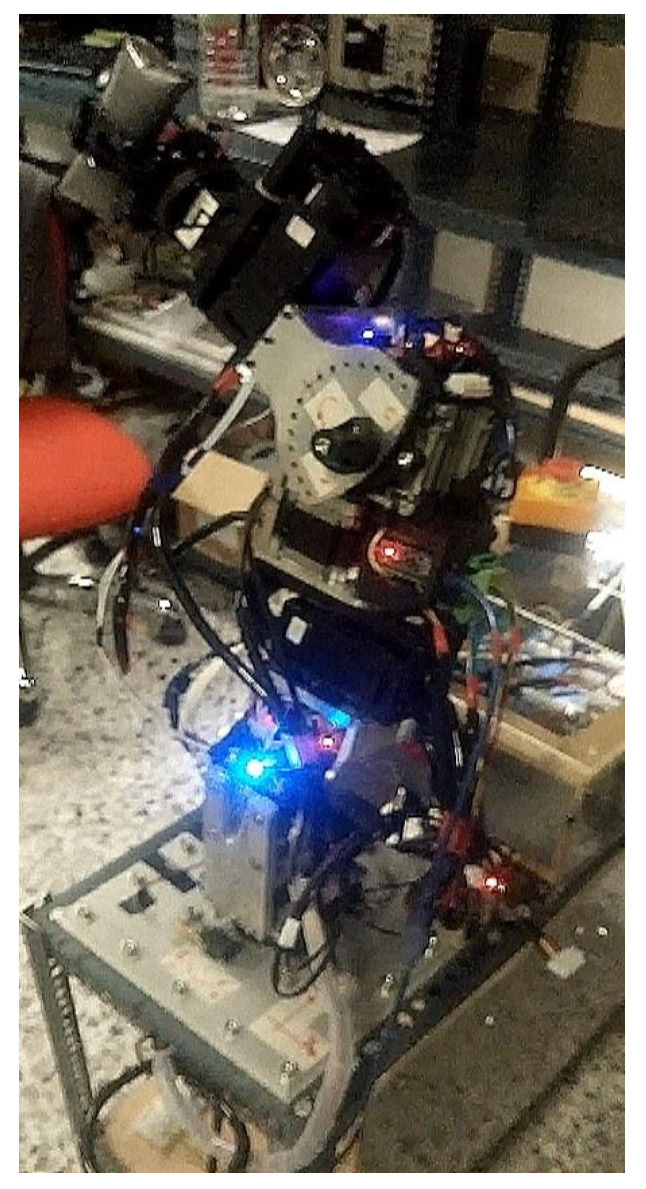
Optimal anatomy of the metamorphic manipulator derived under the MMA measure for Task 1.

**Figure 13 sensors-22-00918-f013:**
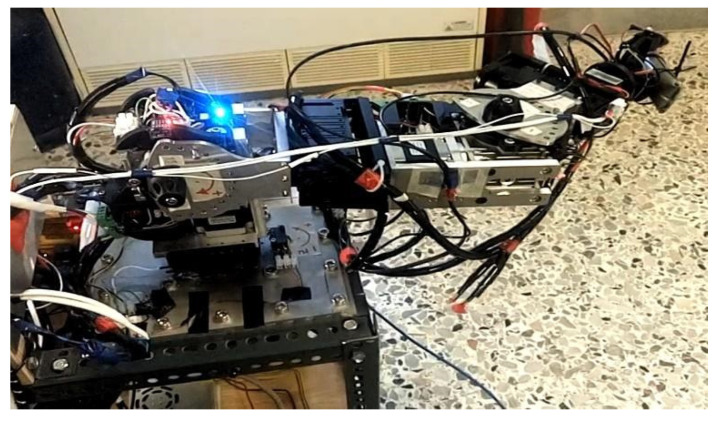
Optimal anatomy of the metamorphic manipulator derived under the MVR measure for Task 1.

**Figure 14 sensors-22-00918-f014:**
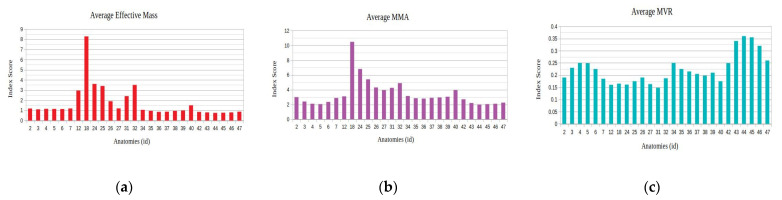
Derived mean index value for the selected performance indices under the available metamorphic structure anatomies, for Task 2 ((**a**) with respect to the Average Effective mass index, (**b**) with respect to the Average MMA index and (**c**) with respect to the average MVR index).

**Figure 15 sensors-22-00918-f015:**
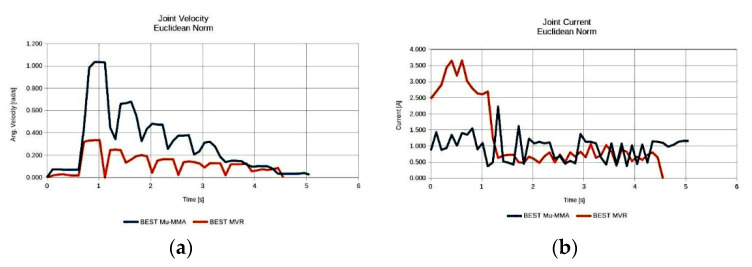
Comparison of the required joint velocities (**a**) and joint current usage (**b**) for the best anatomies under each performance index for Task 2.

**Figure 16 sensors-22-00918-f016:**
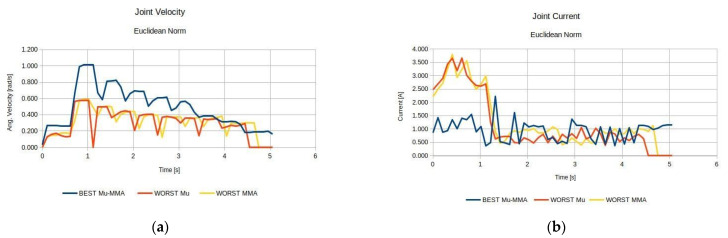
Comparison of the required joint velocities (**a**) and joint current usage (**b**) for the best anatomy and the worst performing anatomy under the MMA and effective mass performance index for Task 2.

**Figure 17 sensors-22-00918-f017:**
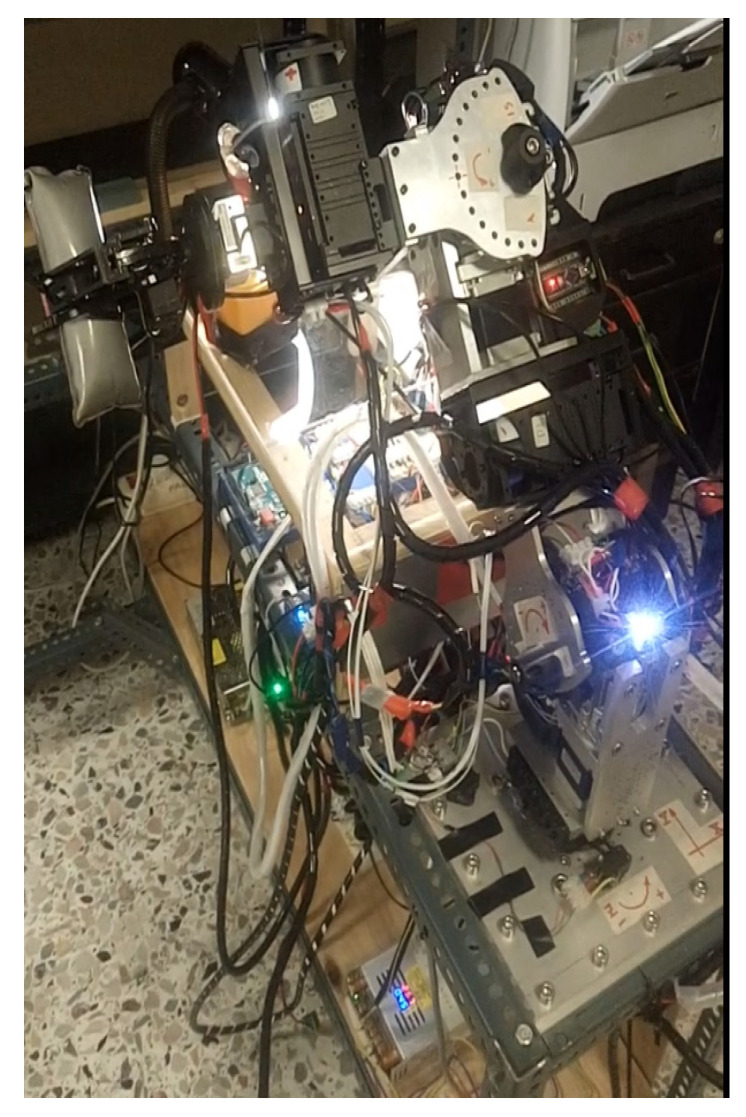
Optimal anatomy of the metamorphic manipulator derived under the effective mass and MMA measure for Task 2.

**Figure 18 sensors-22-00918-f018:**
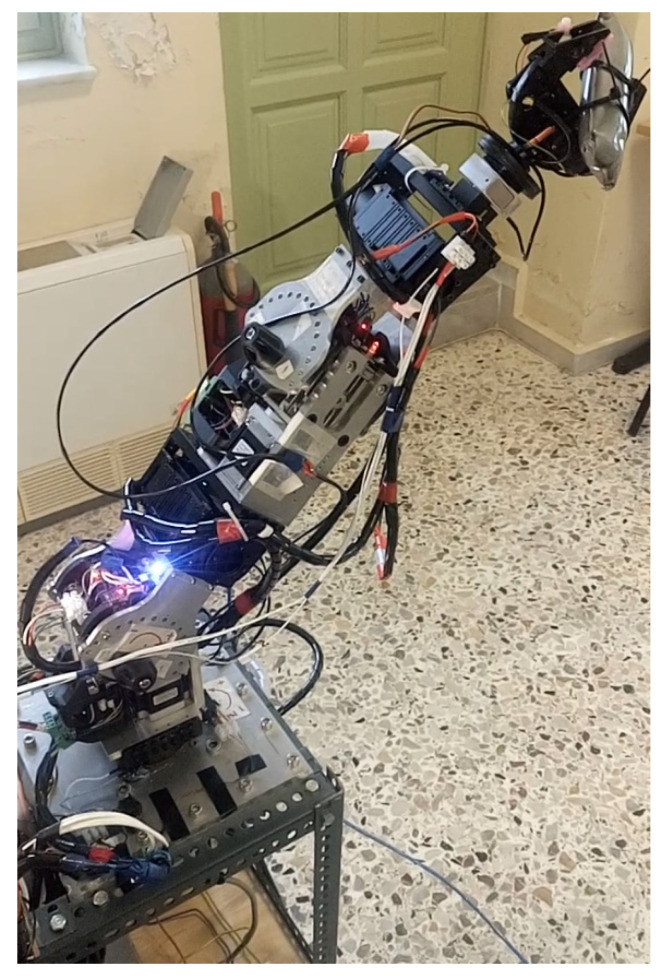
Optimal anatomy of the metamorphic manipulator derived under the MVR measure for Task 2.

**Figure 19 sensors-22-00918-f019:**
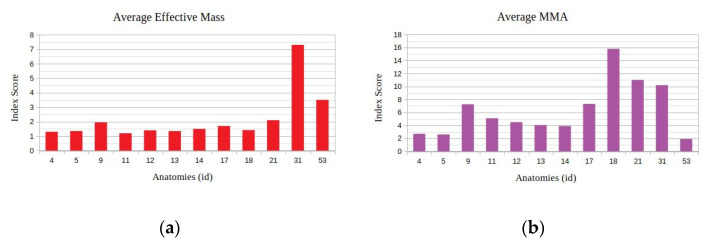
Derived mean index value for the selected performance indices under the available metamorphic structure anatomies, for Task 3 ((**a**) with respect to the Average Effective mass index, (**b**) with respect to the Average MMA index).

**Figure 20 sensors-22-00918-f020:**
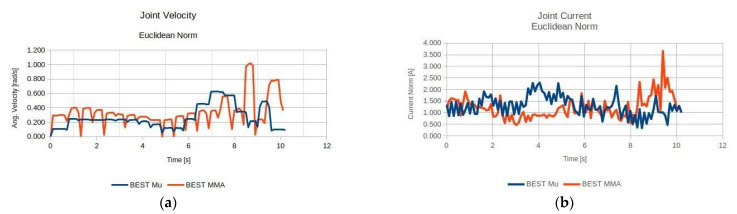
Comparison of the required joint velocities (**a**) and joint current usage (**b**) for the best anatomies under each performance index for Task 3.

**Figure 21 sensors-22-00918-f021:**
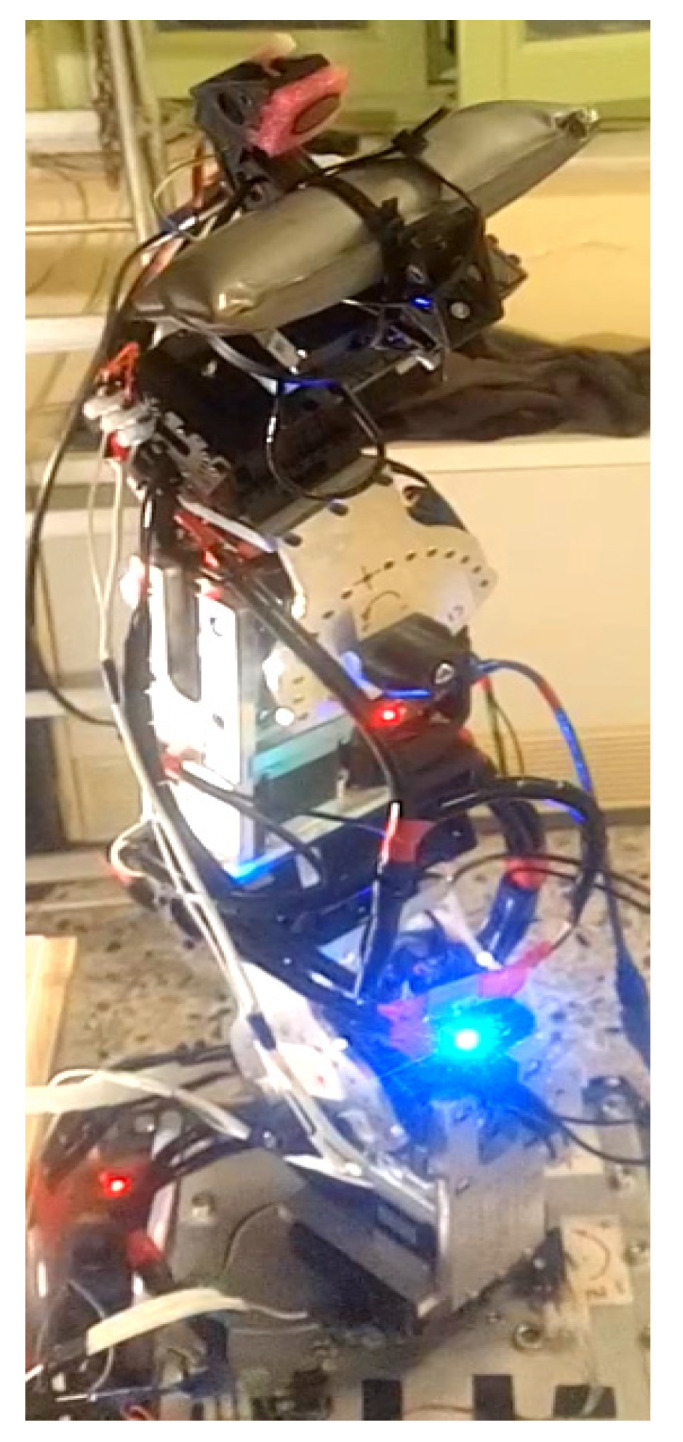
Optimal anatomy of the metamorphic manipulator derived under the effective mass measure for Task 3.

**Figure 22 sensors-22-00918-f022:**
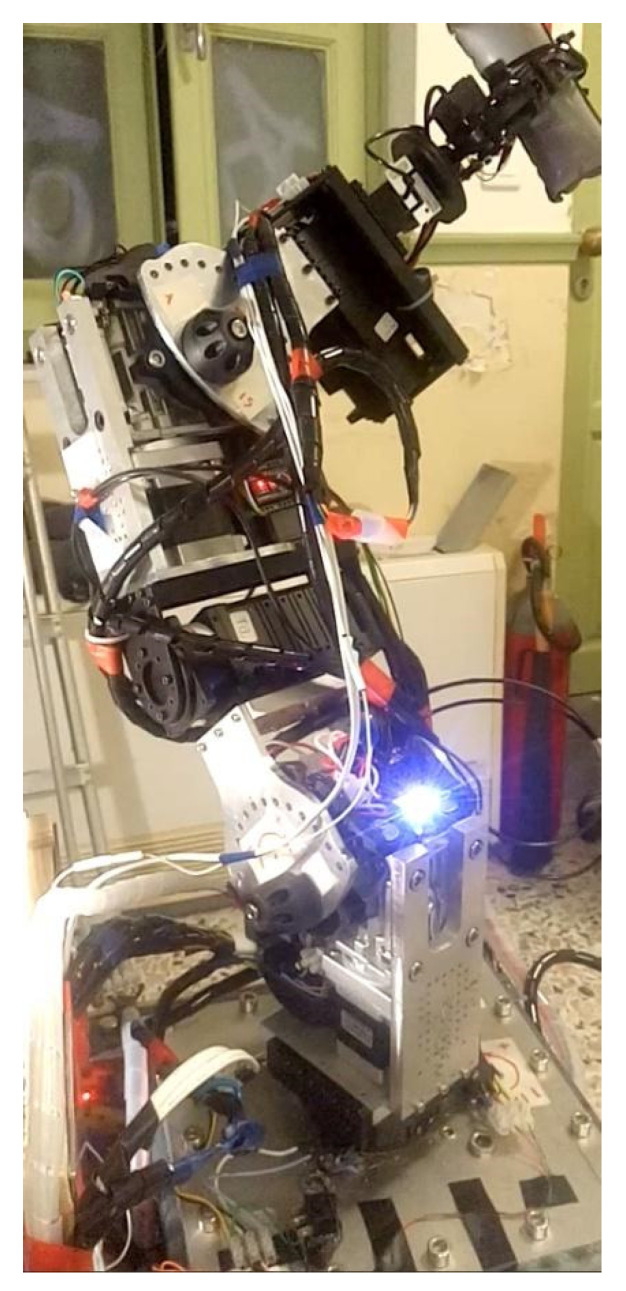
Optimal anatomy of the metamorphic manipulator derived under the MMA measure for Task 3.

**Figure 23 sensors-22-00918-f023:**
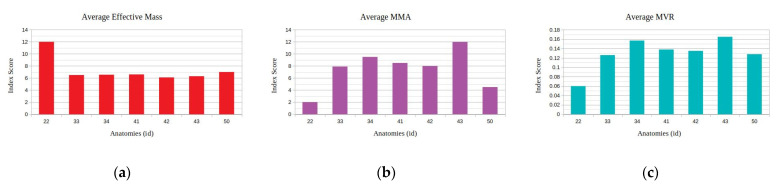
Derived mean index value for the selected performance indices under the available metamorphic structure anatomies, for Task 4 ((**a**) with respect to the Average Effective mass index, (**b**) with respect to the Average MMA index and (**c**) with respect to the average MVR index).

**Figure 24 sensors-22-00918-f024:**
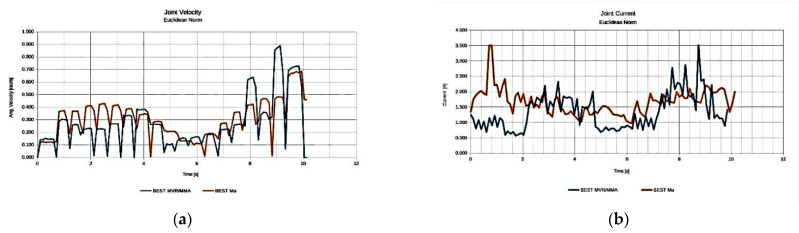
Comparison of the required joint velocities (**a**) and joint current usage (**b**) for the best anatomies under each performance index for Task 4.

**Figure 25 sensors-22-00918-f025:**
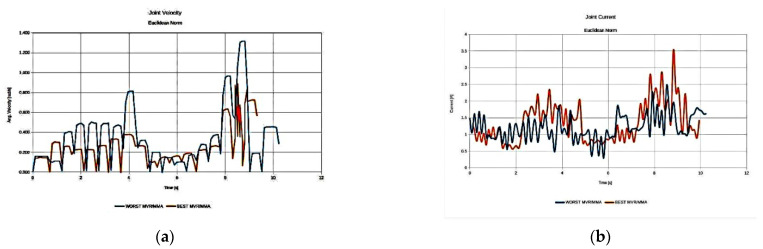
Comparison of the required joint velocities (**a**) and joint current usage (**b**) for the best anatomy and the worst performing anatomy under the MMA and MVR performance index for Task 4.

**Figure 26 sensors-22-00918-f026:**
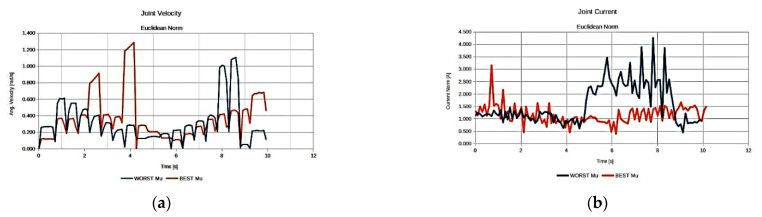
Comparison of the required joint velocities (**a**) and joint current usage (**b**) for the best anatomy and the worst performing anatomy under the Effective Mass performance index for Task 4.

**Figure 27 sensors-22-00918-f027:**
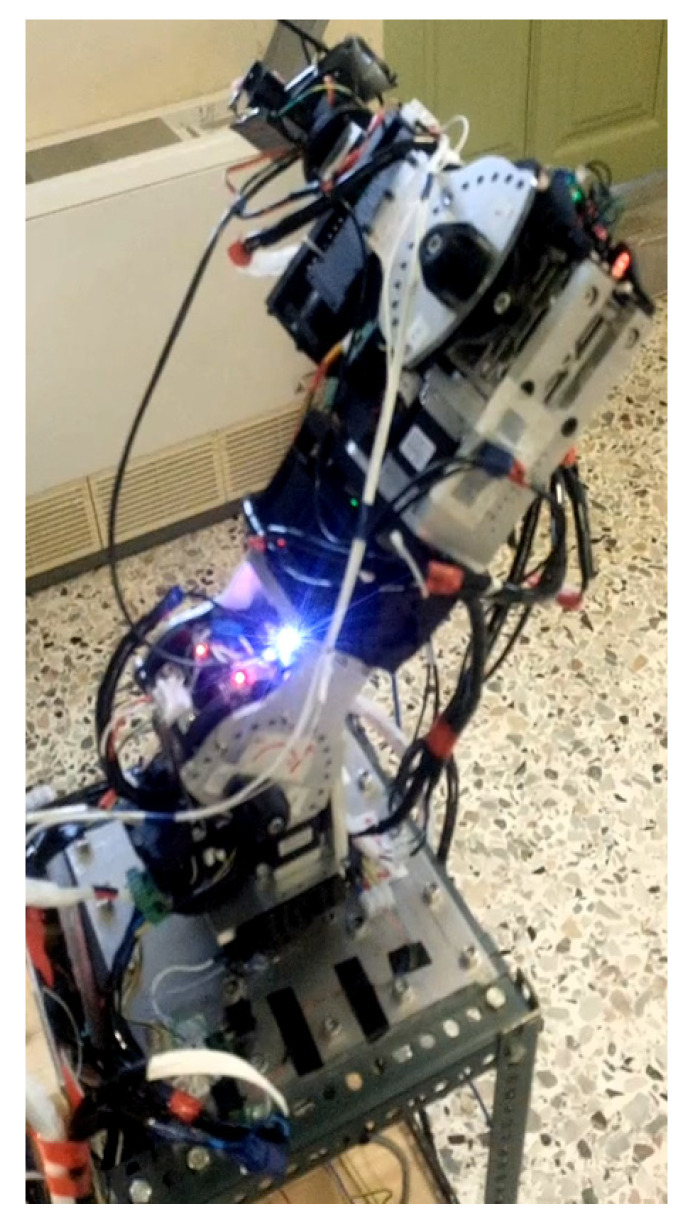
Optimal anatomy of the metamorphic manipulator derived under the effective mass measure for Task 4.

**Figure 28 sensors-22-00918-f028:**
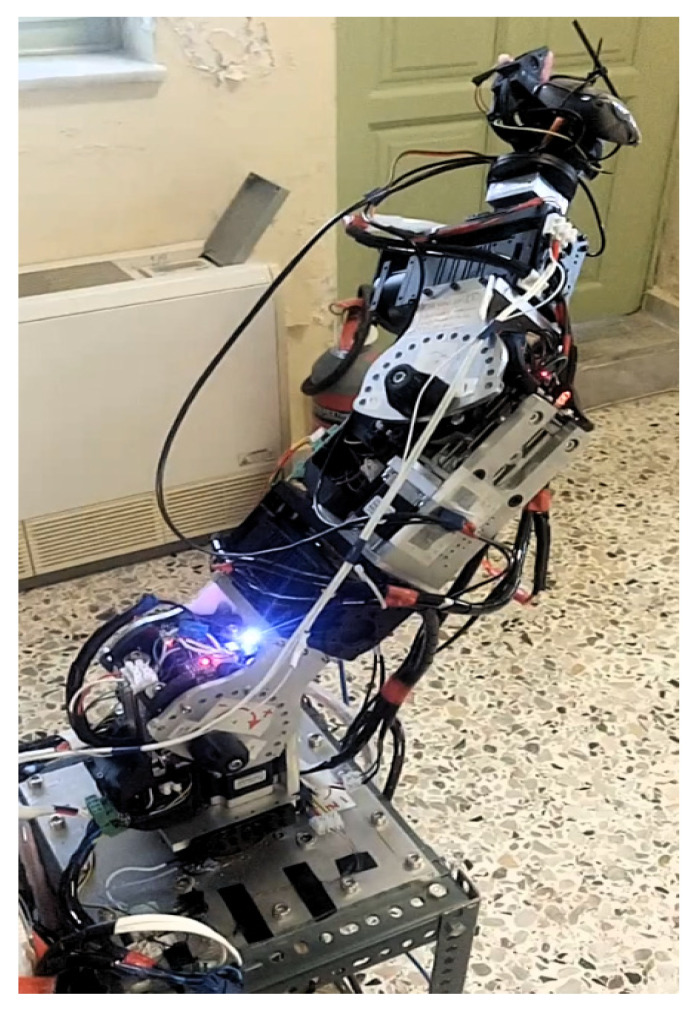
Optimal anatomy of the metamorphic manipulator derived under the MMA and MVR measure for Task 4.

**Table 1 sensors-22-00918-t001:** SMM active joints limit values.

	Joint 1 (AJ1)	Joints 2–3 (AJ2–3)
Angle limit [rad]	2.15	1.60
Velocity limit [rad/s]	1.0	1.57
Acceleration limit [rad/s^2^]	20	87
Torque limit [Nm]	120	60
Current Limit [A]	4	8

**Table 2 sensors-22-00918-t002:** Definition of executed tasks.

Task	Initial Joint Space Point [Rad]	Direction/Rotation Axis	Distance/Radius [cm]
1 Linear path	[−1.0 −0.25 0.50]	+X	20
2 Linear path	[−1.0 −0.25 0.50]	−Z	20
3 Circular path	[−1.0 −0.25 0.50]	+Z	15
4 Circular path	[0.1 0.25 −0.25]	+Z	10

## Data Availability

Data sharing not applicable.

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
