# Peer review of "Experimental Verification of Optimized Anatomies on a Serial Metamorphic Manipulator"

_sensors, 2022, doi:10.3390/s22030918_

Round 1

Reviewer 1 Report

Some of the Reviewer comments given below:

  1. Please correct the typo error in Fig1.b. The pseudo joint mdule and its respective connectivity surfaces.
  2. If all calculated waypoints satisfy the joint bound constraints and no singularities occur along the calculated trajectory points, how?
  3. In figure 7, y axis no label was presented.
  4. In fig.8b, joint current usage of Best MVR is higher than other two method. Why?
  5. How the joint velocity of robot is measured. Explain?
  6. Conclusion part should include values of the results.
  7. How optimal anatomy of the metamorphic manipulator position was achieved?
  8. Section 4, discussion part is very short and lack of scientific discussions.

Reviewer 2 Report

This paper investigates the methodology for an off-line process to determine the optimal anatomy maximizing performance under different requirements. include the kinematic, kinetostatic, and dynamic performance of the manipulator.

The paper is well written; however, I have the following comments:

  1. Could the authors elaborate more on eq 7,8, and 9? In other words, why do the authors use the average in the cost function, and if the average is the most effective approach?
  2. The algorithm presented follows a specific order to determine the optimal solution; the authors need to explain why this specific order and dose the optimal solution will be different if the order changes. An illustrative example perhaps needed to show the advantage of the specific order.
